# Transparent Object Tracking Benchmark

## Reproducibility Summary

**Scope of Reproducibility**

In the article, the authors of the Transparent Object Tracking Benchmark compare the performance of 25 state-of-the-art tracking algorithms, evaluated on the TOTB dataset, with a new proposed algorithm for tracking transparent objects called TransATOM. Authors claim that it outperforms all other state-of-the-art algorithms. They highlight the effectiveness and advantage of transparency feature for transparent object tracking. They also do a qualitative evaluation of each tracking algorithm on various typical challenges such as rotation, scale variation etc.

**Methodology**

In addition to the TransAtom tracker, we chose ten, best performing on TOTB dataset, state-of-the-art tracking algorithms to evaluate on the TOTB dataset using a set of standard evaluation tools. On different sequences, we performed a qualitative evaluation of each tracking algorithm and thoroughly compared the ATOM tracker to the TransATOM tracker. We did not implement the trackers from scratch, but instead used GitHub implementations. TOTB dataset had to be integrated into some of the standard evaluation tools. We used an internal server with an Ubuntu 18.04 operating system and a TITAN X graphics card to reproduce the results.

**Results**

The tracking performance was reproduced in terms of success, precision, and normalized precision, and the reported value is in the 95 percent confidence interval, which supports the paper's conclusion that TransATOM significantly outperforms other state-of-the-art algorithms on TOTB database. Also, it supports a claim that including a transparency feature in the tracker improves performance when tracking transparent objects. However, we refuted the claim that TransATOM well handles all challenges for robust target localization.

**What was easy**

The evaluation of the tracking results and comparison of different trackers with each other was a simple part of the reproduction because the implementation in Matlab is very robust and works for different formats of tracker results.

**What was difficult**

The most difficult aspect of the replication was integrating the TOTB dataset into various standard evaluation tools and running all trackers on this dataset. The reason for this is that each tool requires its own dataset format, and it was also difficult to set up so many different tracker environments. It also took a long time to run all of the trackers because some of them are quite slow and the TOTB dataset is quite large. The deprecation of different packages was also a problem for some trackers, necessitating extensive debugging.

**Communication with original authors**

We communicated with the author via email. The author provided us with feedback that helped us reproduce the results more accurately.

Submitted to ML Reproducibility Challenge Fall 2021. Do not distribute.

# 1 Introduction

In recent years, the tracking community has made amazing progress. Many new tracking methods, particularly neural network trackers, have substantially improved the tracking of opaque objects. Existing research in the topic mostly focuses on tracking of opaque objects, with very little attention dedicated to tracking of transparent objects. However, transparency brings additional challenges not well tackled by the state-of-the-art in opaque object tracking.

Tracking of such objects may be very relevant to robotic vision, human-machine interacation and security surveillance. A vessel collecting plastic from the sea, for example, could so effectively track plastic in the sea and remove it from the water. Another potential application is the grabbing of produced light bulbs with a robotic arm.

That is why it is important to reproduce and verify the results of this article, as it proposes a new state-of-the-art tracker TransAtom, which is currently thought to be the best performing when tracking transparent objects compared to other trackers.

# 2 Scope of reproducibility

In our work, we focused on reproducing and comparing the results of various trackers on TOTB. The proposed TOTB - transparent object tracking benchmark, which is the first benchmark dedicated to transparent object tracking, was the original paper's main contribution. In our opinion, this is a significant contribution because it is the first step toward the development of trackers for transparent objects.

We intended to evaluate and compare the proposed TransATOM tracker to other state-of-the-art trackers on the aforementioned dataset. With this, we hoped to validate the claim that TransATOM significantly outperforms other state-of-the-art trackers designed primarily for tracking opaque objects. We hoped to verify the claim that including a transparency feature in the tracker improves performance when tracking transparent objects. Finally, we conducted a qualitative evaluation on various types of recordings to see where different trackers excel and where they fail.

Claims that we tested are the following:

- TransATOM assessed on TOTB significantly outperforms other evaluated state-of-the-art algorithms by a large margin.
- Including a transpareny feature in the tracker improves performance when tracking transparent objects.
- TransATOM well handles all challenges for robust target localization owing to the transparency features.

# 3 Methodology

We used the author's TransATOM code to reproduce the results, which is available here. Code for evaluation tools and other trackers was obtained from GitHub:

- PyTracking tool, which includes code for ATOM, PrDiMP-18, PrDiMP-50, DiMP-18 and DiMP-50.
- PySOT tool, which includes code for SiamMask, SiamRPN and SiamRPN++.
- py-MDNet tool, which includes code for MDNet.
- STARK tracker with it's own evaluation tool.

We used an internal server with an Ubuntu 18.04 operating system and a TITAN X graphics card with 12GB VRAM to reproduce the results.

## 3.1 Model descriptions

Here we list hyperlinks to the models' descriptions and all of the parameters we used: TransATOM, ATOM, PrDiMP-18, PrDiMP-50, DiMP-18, DiMP-50, SiamMask, SiamRPN, SiamRPN++, MDNet and Stark.

All models have been pre-trained.

## 3.2 Datasets

An important part of tracker analysis is to know in which cases the tracker excels and in which cases fails. To this end, the authors have assigned several attributes to each recording. A twelve-dimensional binary vector was provided for each sequence to indicate the presence of an attribute (1 denotes the presence of a certain attribute). From Table 1, which summarizes TOTB dataset, we can observe number of sequences with a certain attribute (bold diagonal) and number of sequences with combination of different attributes. Table 2 shows the distribution of the following attributes on TOTB dataset: *illumination variation* (IV), *partial occlusion* (PC), *deformation* (DEF), *motion blur* (MB), *rotation* (ROT), *background clutter* (BC), *scale variation* (SV), *full occlusion* (FOC), *fast motion* (FM), *out-of-view* (OV), *low resolution* (LR) and *aspect ratio change* (ARC). The most frequent challenges in TOTB dataset are *rotation*, *partial occlusion* and *scale variation*. The TOTB dataset is available for download at the following link.

Table 1: Summary of statistics of the TOTB dataset.

| Number of videos | 225 | Number of attributes | 12 |
|---|---|---|---|
| Average duration | 12.7 seconds | Object categories | 15 |
| Total frames | 86,000 | Average frames | 381 |
| Max frames | 500 | Min frames | 126 |

Table 2: Distribution of 12 attributes on the TOTB dataset. The diagonal elements corresponds to the distribution over the entire dataset, each row/column presents the joint distribution for the attribute subset. In other words, the diagonal represents the number of sequences with a specific attribute, while the other values represent the number of sequences with a specific combination of attributes.

|  | IV | POC | DEF | MB | ROT | BC | SV | FOC | FM | OV | LR | ARC |
|---|---|---|---|---|---|---|---|---|---|---|---|---|
| IV | **69** | 24 | 7 | 16 | 43 | 5 | 20 | 2 | 10 | 2 | 3 | 16 |
| POC | 24 | **110** | 18 | 38 | 59 | 23 | 48 | 9 | 26 | 7 | 12 | 40 |
| DEF | 7 | 18 | **42** | 6 | 6 | 8 | 24 | 0 | 7 | 0 | 1 | 20 |
| MB | 16 | 38 | 6 | **69** | 50 | 16 | 29 | 7 | 18 | 6 | 5 | 27 |
| ROT | 43 | 59 | 6 | 50 | **123** | 21 | 59 | 7 | 27 | 6 | 9 | 61 |
| BC | 5 | 23 | 8 | 16 | 21 | **42** | 17 | 3 | 5 | 1 | 0 | 11 |
| SV | 20 | 48 | 24 | 29 | 59 | 17 | **95** | 0 | 33 | 0 | 14 | 68 |
| FOC | 2 | 9 | 0 | 7 | 7 | 3 | 0 | **10** | 0 | 3 | 0 | 0 |
| FM | 10 | 26 | 7 | 18 | 27 | 5 | 33 | 0 | **44** | 0 | 11 | 29 |
| OV | 2 | 7 | 0 | 6 | 6 | 1 | 0 | 3 | 0 | **9** | 0 | 0 |
| LR | 3 | 12 | 1 | 5 | 9 | 0 | 14 | 0 | 11 | 0 | **18** | 11 |
| ARC | 16 | 40 | 20 | 27 | 61 | 11 | 68 | 0 | 29 | 0 | 11 | **82** |

## 3.3 Experimental setup and code

For each tracker, we used one-pass evaluation (OPE), using three measures: *precision* (PRE), *normalized precision* (NPRE) and *success* (SUC). *Precision* is defined as the distance between the centers of the bounding boxes (between the groundtruth and the tracking result), with the value varying depending on the threshold (which may be different for each tracker). *Normalized precision* is used to eliminate the influence of different scales by performing normalization with target areas. *Success* compares the intersection over union (IoU) of tracking results and groundtruth boxes, and success score is calculated as the percentage of tracking results with IoU greater than 0.5.

All the code we needed is available on GitHub.

## 3.4 Computational requirements

Table 3 shows the average FPS, maximum FPS, average one pass evaluation (OPE) time and maximum OPE for each tracker. We spent over 85 hours total evaluating all of the trackers on the GPU, as we evaluated each tracker three times.

Table 3: Each tracker's average FPS, maximum FPS, average one pass time and maximum OPE. We run each tracker three times.

| Tracker | TransATOM | ATOM | PrDiMP-18 | PrDiMP-50 | DiMP-18 | DiMP-50 | SiamMask | SiamRPN | SiamRPN++ | MDNet | Stark |
|---|---|---|---|---|---|---|---|---|---|---|---|
| average FPS | 12 | 25 | 7 | 5 | 7 | 6 | 55 | 42 | 40 | 3 | 70 |
| maximum FPS | 21 | 32 | 13 | 11 | 9 | 8 | 78 | 64 | 59 | 5 | 98 |
| average OPE | 1 h 59 min | 57 min | 3 h 25 min | 4 h 45 min | 3 h 25 min | 3 h 58 min | 26 min | 35 min | 35 min | 7 h 57 min | 20 min |
| maximum OPE | 2 h 2 min | 59 min | 3 h 29 min | 4 h 50 min | 3 h 29 min | 4 h 04 min | 28 min | 36 min | 37 min | 8 h 13 min | 21 min |

# 4 Results

There are two parts to this section. In section 4.1, we examine tracker performance in terms of precision, normalized precision and success and compare it with each other. The results presented there are to support the first two claims in Section 2. The results of the qualitative evaluation are presented in section 4.2, where the results contradicts the claim that TransATOM effectively handles all challenges for robust target localization.

## 4.1 Performance results

We compare the performance of the trackers we chose by taking only the trackers with the best evaluation results on the TOTB dataset reported in the original article. Besides that, we evaluated the current state-of-the-art transformer-based tracker Stark. Figure 1 shows the precision plot, normalized precision plot, and success plot of one pass evaluation (OPE) on the TOTB dataset. The average performance measures with standard error are shown in Table 4. When we compare the precision for the TransATOM and SiamRPN++ tracker, we see that the TransATOM has a higher average score. Tracker Stark is unquestionably the best.

The first two positions do not change when we compare normalized precision. SiamMask is the third best tracker in terms of normalized precision, with an average normalized precision of 72.4 percent. The average normalized precision of TransATOM is higher here as well. The average score for Stark is indeed the highest, and the confidence interval does not overlap with any of the other tracker intervals.

When we observe success, the situation change (recall that the success score is calculated as the percentage of tracking results with IoU greater than 0.5). TransATOM has a confidence interval of $[73, 74.6]$, while PrDiMP_50 has a 95% confidence interval of $[72.9, 74.5]$. It is worth noting that the intervals overlap substantially, where TransATOM has a higher lower and higher bound. Because the success score is determined by the threshold, this comparison is less important than the normalized precision score comparison, because the IoU for a particular tracker may be slightly lower than the 0.5 threshold, resulting in a significantly lower success score.

Figure 1: Tracking performance of 10 state-of-the-art trackers and TransAtom in terms of precision, normalized precision and success.

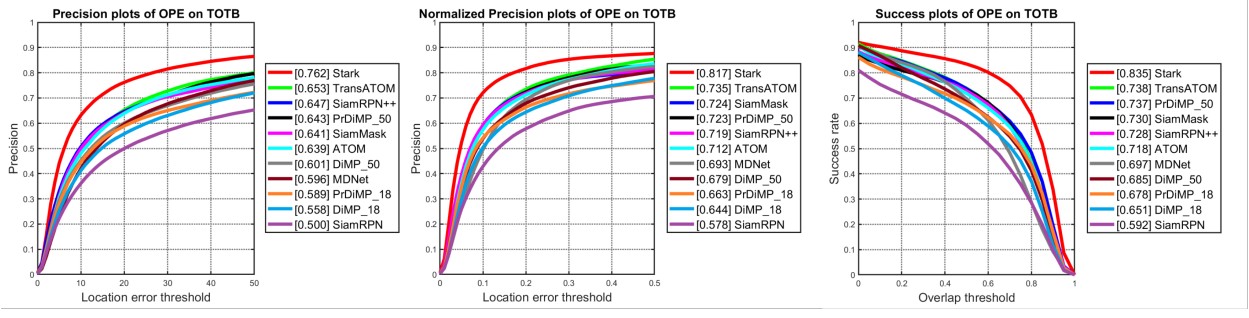

TransATOM and ATOM tracker performance can now be compared. The only distinction between these two trackers is that TransATOM includes a transparency feature. We can confirm the second claim, that including a transparency feature in the tracker improves performance when tracking transparent objects, because TransATOM outperforms ATOM by 1.4 percent in precision, 2.3 percent in normalized precision, and 2.0 percent in success, while confidence intervals do not overlap (see Table 4 and Figure 1).

Table 4: For each tracker, the average precision, normalized precision, and success with a standard error are specified.

| Tracker | Precision | Normalized Precision | Success |
| --- | --- | --- | --- |
| TransATOM | $65.3 \pm 0.4$ | $73.5 \pm 0.5$ | $73.8 \pm 0.4$ |
| ATOM | $63.9 \pm 0.3$ | $71.2 \pm 0.4$ | $71.8 \pm 0.4$ |
| DiMP18 | $55.9 \pm 0.6$ | $64.4 \pm 0.8$ | $65.1 \pm 0.7$ |
| DiMP50 | $60.1 \pm 0.5$ | $67.9 \pm 0.8$ | $68.5 \pm 0.7$ |
| prDiMP18 | $58.9 \pm 0.3$ | $66.3 \pm 0.4$ | $67.8 \pm 0.3$ |
| prDiMP50 | $64.3 \pm 0.3$ | $72.3 \pm 0.4$ | $73.7 \pm 0.4$ |
| SiamRPN | $62.9 \pm 0.2$ | $70.1 \pm 0.4$ | $72.2 \pm 0.4$ |
| SiamMASK | $64.1 \pm 0.3$ | $72.4 \pm 0.4$ | $73.0 \pm 0.3$ |
| SiamRPN++ | $64.7 \pm 0.2$ | $71.9 \pm 0.5$ | $72.8 \pm 0.5$ |
| Stark | $76.2 \pm 0.4$ | $81.7 \pm 0.5$ | $83.5 \pm 0.4$ |
| MDNet | $59.6 \pm 1.8$ | $69.3 \pm 2.9$ | $69.7 \pm 2.2$ |

## 4.2 Qualitative evaluation

In this section we present the results of qualitative evaluation of different trackers. We compared the performance of all trackers on the three most common challenges in the TOTB dataset: *rotation*, *partial occlusion*, and *scale variation*, as authors did in the original article. TransATOM is clearly not the best algorithm for certain challenges, as shown in Figure 2. For rotation, the SiamRPN++ tracker is superior, while PrDiMP_50 is superior for partial occlusion and TransATOM is superior for scale variation.

Figure 2 shows the qualitative results of 11 different trackers in six typical difficult challenges. TransATOM has some issues following the entire *rotating* object in (*WineGlass-7*), as it only follows the lower part of the wine glass. TransATOM tracks the wrong object in the *Bulb-5* sequence. It is alsounable to track an object on *GlassSlab-15* sequence with *aspect ratio change*, as well as *JuggleBubble-1* with *partial occlusion*. It works well on the *ShotGlass-10* with *motion blur*, as well as in the *TransparentAnimal-11* with *scale variation*.

Our findings refute the third claim, which states that TransAtom well handles all challenges for robust target localization owing to the transparency feature.

Figure 2: Tracking performance different tracking algorithms on the three most common attributes in TOTB dataset including *rotation*, *partial occlusion* and *scale variation* using success.

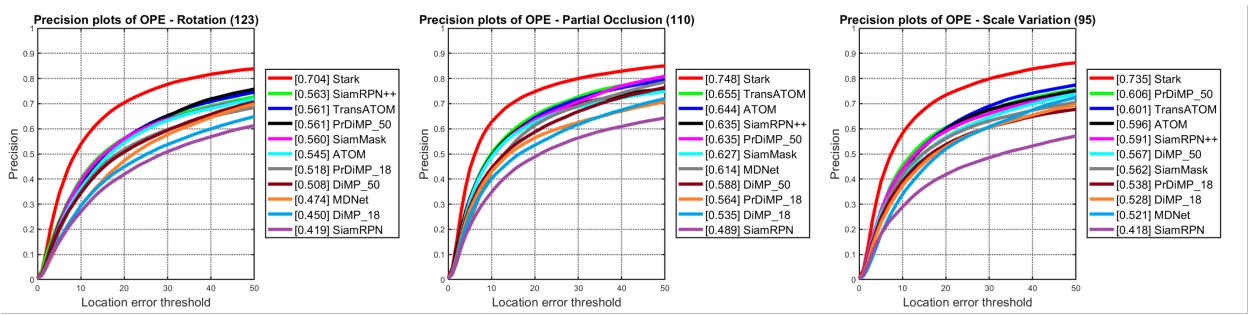

We evaluated the current state-of-the-art tracker Stark in addition to the best trackers from the original article. We can see from the above results that it outperforms all other tracking algorithms and handles all challenges for robust target localization much better.

## 5 Conclusion

We were able to confirm two of the three main claims from the original article, as stated in the results. The claim which states that TransATOM outperforms other evaluated state-of-the-art algorithms by a large margin on TOTB was confirmed, as we demonstrated that TransATOM outperforms other trackers that the authors evaluated in their original

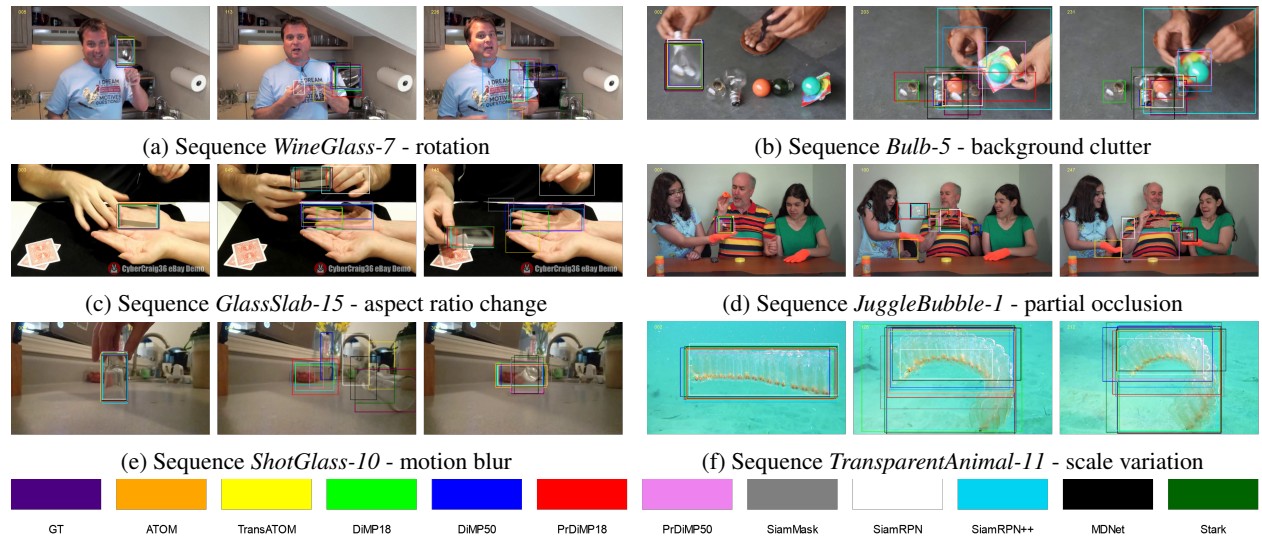

(a) Sequence *WineGlass-7* - rotation

(b) Sequence *Bulb-5* - background clutter

(c) Sequence *GlassSlab-15* - aspect ratio change

(d) Sequence *JuggleBubble-1* - partial occlusion

(e) Sequence *ShotGlass-10* - motion blur

(f) Sequence *TransparentAnimal-11* - scale variation

| GT | ATOM | TransATOM | DiMP18 | DiMP50 | PrDiMP18 | PrDiMP50 | SiamMask | SiamRPN | SiamRPN++ | MDNet | Stark |

Figure 3: Qualitative results of eleven trackers in six typical difficult challenges.

paper. The second claim was also confirmed, because we showed that the difference in performance of TransATOM and ATOM tracker, which differentiate only on the transparency feature, was significant enough.

However, we were unable to confirm the the last claim, which states that TransATOM effectively handles all challenges for robust target localization due to transparency features. We evidenced multiple cases where the TransATOM tracker fails to handle transparent object tracking adequately. We believe that this is the most audacious claim, because we know that the TOTB dataset contains many difficult challenges that no currently-developed tracker can handle well.

The strength of our strategy was that we attempted to follow the steps outlined in the article. We also chose only the top 10 trackers based on their performance on the TOTB dataset, allowing us to focus more on implementation and evaluation quality. In addition, we compared the current state-of-the-art Stark tracker. We wanted to show that there is still a lot of room for improvement in the field of tracking transparent objects. Because we didn't know which parameters were used in the original article, we used only the default choice of parameters for all trackers. This was a flaw in our approach. We could also do more in-depth qualitative analysis because we could compare three results for each tracker and pick the best one, but we took the best one in the whole TOTB dataset.

## 5.1 Recommendations for reproducibility

We recommend using the code from GitHub to reproduce the results of the original article or our work. We recommend to look at which evaluation tool the original code is written in for each tracker and use that evaluation tool. We do not recommend reproducing the results for all trackers, but rather selecting the trackers with the best results, because evaluating the trackers takes a lot of time. We have adapted the TOTB dataset for PySOT, py-MDNet, STARK, and VOT21 evaluation tool, and we recommend to download it from here.

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
