# OpenReview forum: "Transparent Object Tracking Benchmark"
_ML_Reproducibility_Challenge/2021/Fall — RC2021_

### Official Review · Reviewer_AyJ6 · 2022-02-28

**Rating:** 8
**Confidence:** 4

**Review:**

**Scope of reproducibility**

The report defines well the scope of the reproducibility.

**Code**

The authors have re-used public implementations of the different trackers as well as the metrics.

**Communication with original authors**

The authors of the report reached out the authors of the original article, which helped them reproduce the original results.

**Hyperparameter search**

The authors tried to match the original hyperparameters as close as possible.

**Discussion on results**

The authors correctly discuss how the results differ from those originally reported and discuss different justifications for these discrepancies.

**Overall rating**

This report is clear and well written. The authors clearly stated the claims to verify and matched the original experimental setup. Their experiments confirm most of the claims in the original paper, while casting doubt over the claim that TransATOM can handle all challenging tracking scenarios due to its transparency features. Overall this is a great reproducibility effort and I argue for its acceptance.

---

### Official Review · Reviewer_1wer · 2022-03-01
**Review for Transparent Object Tracking Benchmark**

**Rating:** 8
**Confidence:** 4

**Review:**

This paper is well organized and clear. The authors have done a great effort in reproducing the original paper and validating their claims. There are results beyond the original paper. The authors show that TransAtom does not necessarily handle all challenges for robust target localization and support the other two claims of the original paper.
Besides, the authors report that it has been difficult to integrate the TOTB dataset into various standard evaluation tools because each tool requires its own dataset format. Was not this already available in the original code?

Table 4 shows that the Transparent feature increases the performance with respect to ATOM and others. Would not it be valuable to add this transparent feature to the other trackers and see what their performance might look like?

In the paper, it is said that:
"Figure 2: Tracking performance different tracking algorithms on the three most common attributes in TOTB dataset including rotation, partial occlusion and scale variation using success". However, Figure 2 in the paper shows the Precision plots. Is this a mistake?

---

### Official Review · Reviewer_HHtT · 2022-03-16
**Evaluating trackers on a single benchmark**

**Rating:** 7
**Confidence:** 4

**Review:**

1. The authors do a great job of evaluating various trackers, including two concurrent trackers: STARK and TransATOM. It was indeed surprising that Stark which does not explicitly utilize 'transparency features' beats TransATOM which tries to explicitly model transparent objects.
2. The authors however only utilize pre-trained models. It would have been great if the authors could also validate if training the trackers from scratch also results in the reported numbers.

---

### Meta-Review · Area_Chair_mh4S · 2022-04-08

**Recommendation:** Accept
**Confidence:** 4

**Metareview:**

The authors did a very good job at reproducing the results and also provided some additional interesting insights beyond the original paper.  A great reproducibility effort indeed.

---

### Decision · Program_Chairs · 2022-04-09

**Decision:**

Accept

**Comment:**

Following the recommendation of reviewers and meta-reviewer, the paper is accepted for ML Reproducibility Challenge 2021, and will be published in the upcoming special edition of ReScience Journal.